# The Emerging Role of Metformin in the Treatment of Hepatocellular Carcinoma: Is There Any Value in Repurposing Metformin for HCC Immunotherapy?

**DOI:** 10.3390/cancers15123161

**Published:** 2023-06-13

**Authors:** Stavros P. Papadakos, Daniele Ferraro, Gabriele Carbone, Adam Enver Frampton, Giovanni Vennarecci, Stylianos Kykalos, Dimitrios Schizas, Stamatios Theocharis, Nikolaos Machairas

**Affiliations:** 1First Department of Pathology, Medical School, National and Kapodistrian University of Athens, 11527 Athens, Greece; stpap@med.uoa.gr; 2HPB Surgery and Liver Transplant Unit, AORN A. Cardarelli, 80131 Naples, Italy; daniele.ferraro@aocardarelli.it (D.F.); giovanni.vennarecci@aocardarelli.it (G.V.); 3Department of General Surgery and Organ Transplantation, University of Rome “Sapienza”, 00161 Rome, Italy; gaby.carbo@gmail.com; 4Department of Surgery & Cancer, Imperial College London, Hammersmith Hospital, London W12 0NN, UK; a.frampton@imperial.ac.uk; 5Oncology Section, Surrey Cancer Research Institute, Department of Clinical and Experimental Medicine, FHMS, University of Surrey, The Leggett Building, Daphne Jackson Road, Guildford GU2 7WG, UK; 6HPB Surgical Unit, Royal Surrey NHS Foundation Trust, Guildford GU2 7XX, UK; 7Second Department of Propaedeutic Surgery, National and Kapodistrian University of Athens, Laiko General Hospital, 11527 Athens, Greece; skykalos@med.uoa.gr; 8First Department of Surgery, National and Kapodistrian University of Athens, Laiko General Hospital, 11527 Athens, Greece; dschizas@med.uoa.gr

**Keywords:** metformin, hepatocellular carcinoma, diabetes mellitus, antineoplastic, survival, sorafenib, immunotherapy

## Abstract

**Simple Summary:**

The study investigates the potential use of metformin, a medication commonly prescribed for type 2 diabetes, in the treatment of hepatocellular carcinoma (HCC). Metformin has been gaining attention for its possible anticancer properties beyond its antidiabetic effects. The study explores the mechanisms through which metformin may exert its anticancer effects, including its impact on metabolic pathways, inflammation reduction and modulation of the tumor microenvironment. It also discusses the potential of combining metformin with immunotherapeutic approaches, such as immune checkpoint inhibitors, to enhance HCC treatment outcomes. While preliminary preclinical studies have shown promising results, further research and clinical trials are needed to determine the optimal dosage, treatment duration and patient selection criteria for metformin-including therapies. In conclusion, repurposing metformin for HCC immunotherapy presents a hopeful avenue for enhancing treatment options. Utilizing metformin’s immunomodulatory properties in combination with other therapeutic strategies could pave the way for more effective and tailored approaches to HCC treatment in the future.

**Abstract:**

Hepatocellular carcinoma (HCC) is one of the leading causes of cancer-related deaths worldwide. There has been significant progress in understanding the risk factors and epidemiology of HCC during the last few decades, resulting in efficient preventative, diagnostic and treatment strategies. Type 2 diabetes mellitus (T2DM) has been demonstrated to be a major risk factor for developing HCC. Metformin is a widely used hypoglycemic agent for patients with T2DM and has been shown to play a potentially beneficial role in improving the survival of patients with HCC. Experimental and clinical studies evaluating the outcomes of metformin as an antineoplastic drug in the setting of HCC were reviewed. Pre-clinical evidence suggests that metformin may enhance the antitumor effects of immune checkpoint inhibitors (ICIs) and reverse the effector T cells’ exhaustion. However, there is still limited clinical evidence regarding the efficacy of metformin in combination with ICIs for the treatment of HCC. We appraised and analyzed in vitro and animal studies that aimed to elucidate the mechanisms of action of metformin, as well as clinical studies that assessed its impact on the survival of HCC patients.

## 1. Introduction

Accounting for the vast majority of primary hepatic malignancies, hepatocellular carcinoma (HCC) is the sixth-most common newly diagnosed malignancy overall and the third leading cause of cancer-related deaths worldwide [1,2]. Whilst its incidence is anticipated to increase in the future, HCC is acknowledged as a well-known cause of death in cirrhotic patients [1,3]. Substantial progress has been made in understanding the risk factors and epidemiology of HCC, resulting in the implementation of efficient preventative, diagnostic and treatment strategies, which, in turn, have led to a significant impact on survival outcomes [2,4,5,6,7,8]. Chronic hepatitis B and C viral infections, nonalcoholic fatty liver disease and alcoholic cirrhosis constitute the leading causes of HCC, whilst other risk factors such as obesity and type 2 diabetes mellitus (T2DM) are also associated with hepatocellular carcinogenesis [2,9,10]. Of interest, studies have shown that diabetics have a significantly higher chance of developing HCC, regardless of the concurrent presence of chronic viral hepatitis infection, obesity or chronic consumption of alcohol [10,11,12,13].

Metformin is a widely used, first-line, oral hypoglycemic agent for patients with T2DM, which lowers blood glucose by inhibiting hepatic gluconeogenesis and promoting the utilization of glucose in the periphery [14,15]. AMP-activated protein kinase (AMPK) is an enzyme targeted by metformin, which leads to a peripherally increased intake of glucose by the muscles while it also serves as an upstream regulator of LKB1, a protein kinase demonstrated to act as a tumor suppressor [16]. Taking this interplay into account, a study back in 2005 showed a reduced risk of developing cancer in patients with T2DM, and this was potentially due to the administration of metformin [17]. Furthermore, another study published shortly after sought to evaluate the association among antidiabetic regimens and cancer-related mortality in T2DM patients, hypothesizing that the modifications of insulin levels induced by these different agents might have a role in promoting or inhibiting cancer [18]. The authors showed that cancer-related mortality was significantly lower in the metformin group compared to other agents. In that setting, a number of experimental and clinical studies over the last decade have evaluated the antitumor effects of metformin in several malignancies, including HCC [19,20,21]. Despite published outcomes indicating a beneficial role of metformin in patients with HCC, how this protective effect is exactly exerted is not yet clear, and conflicting outcomes have been published. To that end, the objective of our study was to critically review published experimental and clinical studies evaluating the role of metformin in the treatment of HCC.

## 2. Mechanism of Action

Metformin was introduced into clinical practice in 1957. It is a dimethyl biguanide, a synthetic derivative of galegine (isoprenylguanidine), a molecule derived from the plant Galega officinalis, and found to be too toxic when used as a glucose-lowering agent [22]. Despite being used for the last six decades, its mechanism of action is still largely unknown. It has been demonstrated that metformin exerts its actions in the liver, the intestines and systemically on the gut microbiota. It also affects the inflammatory process, aging and the gut–brain–liver interorgan communication to lower the endogenous glucose production (EGP) [22,23].

In the liver, it mainly inhibits gluconeogenesis. In hepatocytes, organic cation transporter-1 (OCT-1) is responsible for metformin uptake (Figure 1) [19]. MATE1 (Multidrug and Toxin Extrusion 1) and MATE2K (Multidrug and Toxin Extrusion 2K) are key transporters that play a crucial role in the pharmacokinetics of metformin [24]. These transporters are responsible for the renal excretion of metformin, thereby influencing its concentration in the body. MATE1 is predominantly expressed in the kidneys, where it is localized on the luminal membrane of proximal tubule cells. It facilitates the efflux of metformin from renal tubular cells into urine, thereby reducing its plasma concentration. Genetic variations in MATE1 have been associated with altered metformin responses and pharmacokinetics [25]. On the other hand, MATE2K, also expressed in the kidneys, is involved in the reabsorption of metformin from urine back into the renal tubular cells. This transporter works in tandem with MATE1 to maintain a delicate balance between the renal excretion and reabsorption of metformin [24]. The interplay between MATE1 and MATE2K in the renal handling of metformin significantly impacts its pharmacokinetics. Genetic variations, drug–drug interactions and the disease states affecting these transporters can influence the disposition of metformin and, consequently, its therapeutic effectiveness [26]. Metformin is subsequently accumulated in cells, concentrated mainly in the mitochondria (the concentration is 1000-fold higher than in the extracellular medium) [19], as its positive charge (NH3+) brings it towards the negative side of the transmembrane potential. In the mitochondria, it inhibits complex I of the respiratory chain, therefore reducing the production of ATP, which is necessary for gluconeogenesis, and consequently, it lowers the ATP/AMP and ATP/ADP ratios [27,28,29]. Moreover, it has been found to possess the ability to inhibit the mitochondrial isoform of glycerophosphate dehydrogenase, a key component of the shuttle responsible for the translocation of reducing equivalents from the cytosol into the mitochondrion for reoxidation. This can cause an impairment of the NAD/NADH+ ratio that contributes to the negative effect of metformin on gluconeogenesis [30]. The reduction in the ATP/AMP and ATP/ADP ratios, an index of the impairment of a cell’s energy balance, activates the energy cell’s sensor AMP-activated protein kinase (AMPK), which, in an attempt to restore energy homeostasis, will switch on catabolic pathways to restore the ATP reserves and inactivate ATP-consuming reactions [31]. Low doses of metformin can also interact with Axin and LAMTOR1 (late endosomal/lysosomal adaptor, MAPK and mTOR activator), a lysosomal protein that, in turn, can activate AMPK through an alternative way [32]. Activated AMPK additionally phosphorylates and activates cAMP-specific 3′, 5′-cyclic phosphodiesterase 4B, hence causing a reduction in the cytosolic level of cAMP, a renowned messenger and intracellular effector. The rise in the AMP/ATP ratio inhibits the conversion of fructose-1, 6-biphosphate into fructose-6-phosphate by fructose-1,6-bisphosphatase, resulting in the acute inhibition of gluconeogenesis and adenylate cyclase, thus contributing to the antagonization of glucagon-induced hyperglycemia obtained by switching from glycolysis to gluconeogenesis via protein kinase A (PKA) [33].

One significant long-term clinically relevant effect of metformin is that it enhances hepatic insulin sensitivity and reduces hepatocytes lipid stores through the AMPK-mediated inhibition of fat synthesis and activation of hepatic fat oxidation by direct phosphorylation of the two isoforms of acetyl-CoA carboxylase (ACC1/ACC2) [33].

The glucose-decreasing impact of metformin can be only partly explained by its action in the liver [34]. In fact, genetic studies conducted in humans with loss-of-function of OCT-1 showed a drastic reduction in liver uptake of the drug without impairing its effect on the reduction of HbA1c in T2DM individuals [35]. Moreover, the modified release formulation of metformin, which has minimal systemic absorption and is largely retained in the gut, is as effective as immediate release metformin in lowering the hepatic levels of glucose in individuals with T2DM [36]. FDG-PET studies have shown that metformin concentrates in colonic enterocytes with the minimal luminal concentration [37]. Part of the gut-mediated metformin effect is due to the increase in production of glucagon-like peptide 1 (GLP-1), leading to an increase in glucose-stimulated insulin secretion and suppressed glucagon secretion [38]. In addition, the stimulation of the vagal afferent fibers in the duodenum (towards nucleus tractus solitarius) could decrease hepatic glucose production via vagal efferent stimulation (gut–brain–liver crosstalk) [39]. Finally, metformin has been proven to modify the gut microbiome with an increase in Escherichia spp. and a decrease in Intestinibacter spp. and to also modulate the inflammatory response via modification of the NF-kB pathway, reducing the differentiation of monocytes into macrophages and reducing the levels of CCL11, a chemokine associated with the tissue aging process [40]. The reduction of the neutrophil-to-lymphocyte ratio (NLR) could potentially lead to a modification of the cardiovascular risk (NLR has been proven to be a risk factor for all-cause death and cardiac events) [41].

Taking the above into consideration, several potential mechanisms of action of metformin in HCC have been proposed based on preclinical and clinical studies [42], including the activation of AMPK [43]; the inhibition of mitochondrial complex I, which disrupts cellular energy metabolism, impairing cancer cell growth and survival [44]; the modulation of insulin signaling, which indirectly influences cancer development and progression [45]; the reduction of inflammation [43] and the effects on cancer stem cells (CSCs) [46]. The orchestration of immune cell function by metformin in the context of HCC involves influencing the metabolic reprogramming of immune cells, particularly T cells, within the tumor microenvironment [47]. The metabolic reprogramming of T cells by metformin involves the activation of AMPK, leading to various metabolic changes, such as an increased glucose uptake, enhanced mitochondrial function and altered fatty acid metabolism [48]. These metabolic alterations can provide T cells with the necessary energy and resources to carry out their effector functions efficiently [48]. Furthermore, metformin has been found to inhibit the function of regulatory T cells (Tregs). Tregs are a specialized subset of T cells that regulate immune responses by suppressing the activity of other immune cells, including effector T cells. In HCC, Tregs can contribute to an immunosuppressive microenvironment, dampening the antitumor immune response. By inhibiting Treg functions, metformin may counteract their immunosuppressive effects [49]. In summary, metformin’s modulation of immune cell functions in HCC involves enhancing the differentiation and function of effector T cells while inhibiting the suppressive activity of regulatory T cells. By promoting the antitumor activity of effector T cells and reducing immunosuppression, metformin can contribute to an improved antitumor immune response in HCC. However, it is important to note that further research is needed to fully understand the specific mechanisms and clinical implications of these effects.

## 3. Clinical Significance of Metformin

Type 2 diabetes mellitus (DM) increases the risk of HCC three-fold [50]. The main molecular feature of metabolic syndrome is insulin resistance, characterized by a reduced sensitivity to insulin and increased insulin secretion [50]. Insulin resistance and high insulin levels contribute to hepatocarcinogenesis through pathways such as PTEN/P13K/Akt and MAPKK. Diabetes negatively affects the prognosis and clinical course of HCC patients, regardless of the underlying cirrhosis cause. Insulin-sensitizing drugs, such as metformin, are known to reduce HCC incidence [50]. Several drug classes targeting manifestations of metabolic syndrome influence the HCC risk. The use of statins has been shown to have a positive chemopreventive impact on the development of HCC. This beneficial effect is influenced by the dosage and is particularly notable in the case of lipophilic statins [51]. Recently, there have been several observational studies and meta-analyses suggesting that metformin use is associated with a reduced risk of developing HCC in patients with diabetes [52]. In addition, some studies have suggested that metformin use may improve the outcomes in patients with HCC who are receiving curative treatment, such as surgical resection or liver transplantation. In a recent study conducted by the Veterans Administration, Kramer et al. identified 85,963 patients with NAFLD and DM. Five hundred and twenty-four patients were diagnosed with HCC on average after 10.3 years of follow-up. Metformin was associated with a 20% lower risk of HCC compared to no medication, while insulin had no effect on the HCC risk. The combination of insulin with other oral medications was linked to an increased risk of HCC by 1.6 to 1.7 times. Adequate glycemic control was associated with a 31% lower risk of HCC [52]. Additionally, in a retrospective multicenter cohort study, 1566 patients with unresectable HCC who were treated with sorafenib in nine tertiary centers in Korea were included. Its primary objective was to identify the prognostic factors that affect long-term survival. The patients were most commonly males (83.8%) suffering from chronic hepatitis B (77.3%) and were categorized as Barcelona Clinic Liver Cancer stage C (BCLC-C) (78.3%). Patients receiving metformin enjoyed an improved long-term survival (adjusted hazard ratio (aHR) = 3.464; *p* < 0.001), thus highlighting the critical synergy of metformin in conjunction with sorafenib [53]. Another phase I study, which evaluated the dose de-escalation of sorafenib with metformin and atorvastatin in HCC, has been conducted. Sorafenib has several side effects, and its high cost limits its use in many countries. The importance of this study lies in its potential to provide a new treatment option for patients with HCC that is more affordable, less toxic and potentially more effective than sorafenib alone [54].

### 3.1. Meta-Analyses about Metformin in HCC

Several meta-analyses have been conducted recently analyzing the effects of metformin in the overall survival (OS) of DM patients with HCC [20], the HCC risk [55], the OS of DM patients after hepatectomy with a curative intent [56], the HCC risk in patients also receiving aspirin [57] and the HCC risk in patients with chronic hepatitis B infection [58].

Zhou et al. conducted an important study that examined the potential benefit of antihyperglycemic agents in the treatment of HCC in patients with type 2 diabetes. The study analyzed data from 12 observational studies, which included 9132 HCC patients with type 2 diabetes. The main objective of the study was to assess the impact of metformin and other antihyperglycemic agents on the OS and disease-free survival (DFS) of HCC patients with type 2 diabetes. They demonstrated that the use of metformin was associated with a 30% reduction in the risk of death (HR 0.70, 95% CI 0.61–0.81, *p* < 0.001) and a 35% reduction in the risk of recurrence (HR 0.65, 95% CI 0.55–0.78, *p* < 0.001). Other antihyperglycemic agents, such as sulfonylureas, thiazolidinediones and insulin, were also found to have a similar beneficial effect on the OS and DFS. The findings of the study indicated that the use of metformin and other antihyperglycemic agents is associated with an improved OS and DFS in HCC patients with type 2 diabetes mellitus (T2DM). The study had several strengths, including the large sample size and the use of multiple databases, that increased the generalizability of the findings [20]. In the same way, Li et al. conducted a meta-analysis that aimed to clarify the association between metformin use and HCC risk and survival amongst individuals with DM. They analyzed 24 studies consisting of 9 case–control studies with 248,433 participants and 15 cohort studies with 1,203,832 participants to evaluate the effect of metformin on the risk of HCC in patients with DM and 9 cohorts with 11,375 patients to investigate the impact of metformin on the mortality of HCC in DM. The results showed that metformin was associated with a decreased risk of HCC in individuals with DM, as evidenced by a random effects model (OR/RR = 0.59, 95% CI 0.51–0.68, I2 = 96.5%, *p* < 0.001). The study also found that metformin use was associated with a decreased all-cause mortality from HCC in individuals with DM (HR = 0.74, 95% CI 0.66–0.83, I2 = 49.6%, *p* = 0.037). They concluded that their findings supported the use of metformin as a protective factor for HCC in individuals with DM [59]. In the same way, Memel et al. investigated the association between aspirin use and the incidence of HCC. The study involved 2,389,019 participants and identified 20,479 cases of incident HCC. The results indicated that the use of aspirin was linked to a significant reduction in HCC risk (adjusted RR, 0.61; 95% CI: 0.51–0.73). Upon conducting subgroup analyses, it was found that the benefits associated with aspirin were more pronounced in studies that adjusted for the concurrent use of statin and/or metformin (RR, 0.45; 95% CI: 0.28–0.64) compared to those that did not [57]. The above indirectly suggests that metformin might have a positive influence on the reduction of HCC risk.

Yuan et al. evaluated the impact of metformin on the prognosis of patients with HCC after curative therapy, including surgical resection, radiofrequency ablation or percutaneous ethanol injection, and analyzed the OS and recurrence-free survival (RFS). Three of the studies included in the analysis were conducted in China and South Korea. The metformin group consisted of 2313 patients with HCC and T2DM who were treated with metformin for a minimum of 12 months. On the other hand, the non-metformin group consisted of 3623 patients who were treated with other antihyperglycemic agents or metformin for less than 12 months. According to the findings, the use of metformin was associated with a significant increase in the OS at 3 years (OR = 1.50, 95% CI: 1.22–1.83, *p* = 0.000) and 5 years (OR = 1.88, 95% CI: 1.47–2.41, *p* = 0.000). Additionally, the use of metformin was associated with a decrease in the recurrence rate at 1 year (OR = 1.31, 95% CI: 1.08–1.59, *p* = 0.007), 3 years (OR = 1.88, 95% CI: 1.48–2.37, *p* = 0.000) and 5 years (OR = 1.83, 95% CI: 1.40–2.40, *p* = 0.000). In conclusion, the study suggested that metformin can significantly improve the OS and decrease the recurrence rate in patients with HCC and T2DM who have undergone curative therapy for HCC [56]. Finally, Campbell et al. assessed the risk factors associated with the progression to HCC in individuals with chronic HBV infection [58]. They identified 68 relevant studies that met the inclusion criteria, which included a total of 25,447 cases of HCC and 576,792 participants with chronic HBV infection. The results indicated that several factors were significantly associated with an increased risk of HCC in individuals with chronic HBV infection. These factors included the male gender; older age; a high viral load; cirrhosis and a family history of HCC and comorbidities such as diabetes mellitus, hypertension and nonalcoholic fatty liver disease (NAFLD). In studies that took metformin use into account, the association between DM and a higher HCC risk became less evident, as patients with DM had a 16% higher risk of developing HCC compared to non-DM individuals (random effects HR 1.16, 95% CI: 1.04–1.29) in the analysis limited to adjusted studies for metformin [58].

On the contrary, Zeng et al. aimed to evaluate the chemopreventive effects of statins, aspirin and metformin on HCC [55]. The study included 25 studies with a total of 2,863,571 participants. As regards metformin use, they included three studies with 125,458 participants. Metformin use did not appear to be associated with a reduced risk of HCC (HR: 0.57; 95% CI: 0.31–1.06) [55]. The above are summarized in Table 1.

### 3.2. Clinical Studies for Metformin in HCC Risk in Diabetes, NAFLD and Viral Hepatitis

Valuable research has been published in recent years about the effects of metformin in HCC in several liver diseases. Valenti et al. conducted a landmark study that investigated the impact of a sustained virological response (SVR) after direct-acting antiviral (DAA) treatment on the metabolic parameters and clinical outcomes in patients with chronic hepatitis C virus (HCV) infection. The study was conducted in Italy and included 7007 patients. Seven hundred and forty-eight (10.7%) patients with chronic HCV infection were followed with liver stiffness measurement (LSM) at 24 weeks. Out of 4578 patients who had advanced fibrosis, 2946 individuals had follow-up data at 24 weeks and 1905 individuals had follow-up data at 24 months after treatment. SVR was attained by 6849 participants, corresponding to a success rate of 97.7%. They demonstrated that diabetes was independently associated with less significant LSM improvement following SVR (*p* = 0.001). Moreover, among patients with advanced fibrosis, diabetes was identified as an independent predictor of de novo HCC (hazard ratio (HR) 2.09, CI: 1.20–3.63, *p* = 0.009) and cardiovascular events (HR 2.73, CI: 1.16–6.43, *p* = 0.021). The use of metformin was found to have a protective effect against HCC (HR 0.32, CI: 0.11–0.96, *p* = 0.043), which remained significant even after adjusting for the propensity score (*p* = 0.038). According to the study, a multidisciplinary management approach for treating metabolic comorbidities may enhance cardiovascular and, potentially, liver-related outcomes through optimization [60]. In the same way, Tsai et al. aimed to investigate whether metformin could lower the risk of HCC in individuals with chronic hepatitis C (CHC) and DM after antiviral therapy [61]. The study enrolled individuals with CHC who had achieved a SVR after interferon-based therapy in Taiwan. Among the enrolled individuals, 10.8% had DM and 82.8% of them were metformin users. During a median follow-up of 4.4 years, 227 patients were diagnosed with HCC. The 5-year cumulative incidence of HCC was 10.9% for non-metformin users and 2.6% for metformin users, whereas individuals without DM had a 3.0% incidence rate (adjusted hazard ratio (aHR) 2.83, 95% CI: 1.57–5.08 and aHR 1.46, 95% CI: 0.98–2.19, respectively). Cirrhosis was the most significant factor associated with a higher risk of HCC, followed by non-metformin DM, older age, male sex and obesity, whereas hyperlipidemia with statins was associated with a lower risk of HCC [61]. Analogously, Vilar-Gomez et al. investigated the association between T2D and metformin use on the outcomes of patients with nonalcoholic steatohepatitis (NASH)-related cirrhosis. A group of 458 patients who were diagnosed with advanced NASH (either with septal/bridging fibrosis or compensated cirrhosis) were tracked in six tertiary hospitals across four regions (Spain, Australia, Hong Kong and Cuba) over a 10-year period. The group showed that patients administered metformin at baseline had a higher cumulative chance of transplant-free survival (*p* < 0.01) and a decreased cumulative incidence of hepatic decompensation (*p* = 0.047). Nevertheless, the incidence of HCC was similar between baseline metformin users and nonusers. A multivariable analysis confirmed that metformin was independently associated with a lower risk of all-cause mortality (HR 0.41; 95% CI: 0.26–0.45; *p* < 0.001), hepatic decompensation (sHR 0.80; 95% CI: 0.74–0.97; *p* = 0.005) and the development of HCC (sHR 0.78; 95% CI: 0.69–0.96; *p* = 0.047). The beneficial effect of metformin administration on transplant-free survival was mainly observed in patients with a HbA1c of <7% and >7% (adjusted HR 0.93; 95% CI: 0.89–0.98 and adjusted HR 0.87; 95% CI: 0.84–0.90), but the association with hepatic decompensation and HCC was seen only in patients with HbA1c >7% (adjusted sHR 0.97; 95% CI: 0.95–0.99 and adjusted sHR 0.67; 95% CI: 0.45–0.94) [62]. Conclusively, a cohort of patients with biopsy-confirmed NASH and Child–Pugh class A cirrhosis from multiple countries indicated that T2D was associated with an elevated risk of mortality and liver-related outcomes such as HCC. Contrarily, patients who received metformin had a greater probability of survival and a decreased likelihood of developing decompensation or HCC [62]. Table 2 presents selectively clinical studies about the effects of metformin accessing HCC risk in diabetes, NAFLD and viral hepatitis.

### 3.3. Clinical Studies about Metformin Use after Primary Treatment

A significant number of clinical studies from Asian and European centers have sought to evaluate the potential survival benefits induced by the administration of metformin following the primary treatment of HCC, including ablation, radiotherapy, surgical resection or transarterial chemoembolization (TACE). The above are summarized in Table 3.

## 4. Preclinical Evidence about Metformin Influence on HCC Hallmarks

Preclinical evidence suggests that metformin may have potential in the prevention and treatment of HCC by targeting several cancer hallmarks, such as proliferation and apoptosis, the avoidance of drug resistance, autophagy, angiogenesis, metastasis and epigenetic regulation. These effects will be analyzed below.

### 4.1. The Effects of Metformin on HCC Proliferation and Apoptosis

Preliminary studies during the past decade have investigated the effects of metformin on proliferation and apoptosis based on clinical studies stating that AMPK activation was significantly correlated with a better overall survival and disease-free survival in HCC patients [75]. Analogously, a retrospective cohort study investigated the potential role of metformin in reducing the risk of developing HCC in diabetic patients with chronic liver disease. The utilization of metformin was linked to a notable decrease in the likelihood of HCC incidence. The risk reduction was dose-dependent, with higher doses of metformin associated with a greater risk reduction [76]. Unarguably, the ability of a cell to preserve proliferative signaling and avoid apoptosis are the most fundamental characteristics of cancer cells [77]. Mounting evidence suggests that metformin regulates proliferation and apoptosis signaling in HCC, interacting with a multitude of signaling pathways.

Li et al. utilized a transcriptome analysis to explore the underlying mechanisms of metformin’s anticancer effects. The study used the Huh-7 cell line treated with metformin at a concentration of 5 mM for 48 h, and RNA was extracted for the transcriptome analysis. The analysis identified a total of 443 differentially expressed genes, with 237 upregulated and 206 downregulated genes, in the metformin-treated cells compared to the control group. They reported that metformin treatment led to significant changes in the expression of genes related to cell cycle regulation, apoptosis and metabolism. The expression of genes involved in the G1/S transition of the cell cycle, such as *CDK4* and *CCND1*, was downregulated, suggesting that metformin treatment may inhibit cell proliferation. Additionally, the expression of genes related to apoptosis, such as *BCL2L1* and *MCL1*, was downregulated, indicating that metformin may induce cell death [78]. The study also found that metformin treatment resulted in changes in the expression of genes involved in metabolic processes, such as the downregulation of genes related to fatty acid synthesis and the upregulation of genes involved in fatty acid oxidation [78]. Zheng et al. showed both in vitro and in vivo that the activation of AMPK by metformin suppressed NF-kB/STAT3 signaling, leading to the inhibition of HCC cell growth [75], while Cai et al. found that metformin treatment inhibited the growth and proliferation of HepG2 and PLC/PRF/5 cells in vitro and of nude mice in vivo through the induction of cell cycle G1/G0 phase arrest and the upregulation of the tumor suppressor proteins p21CIP and p27KIP. In addition, they found that metformin treatment downregulated the expression of cyclin D1, a key regulator of the cell cycle [79]. Miyoshi et al. treated the HepG2 and Huh7 cell lines with metformin and assessed its effects on cell viability, cell cycle progression and apoptosis. They showed that metformin significantly inhibited cell growth and induced G0/G1 cell cycle arrest in both cell lines. Additionally, metformin increased the expression of the cyclin-dependent kinase inhibitors p21 and p27, which are involved in cell cycle arrest, and decreased the expression of the cell cycle regulator cyclin D1 [80]. Furthermore, metformin induced apoptosis, as evidenced by an increase in caspase 3 activity and PARP cleavage, which are markers of apoptosis [80]. Taking it a step further, Cauchy et al. demonstrated in Alex, HLE and Huh7 cells that the metformin-induced inhibition of mTOR signaling was associated with an increased activation of the PI3K/Akt and Ras/ERK pathways. Both the in vitro and in vivo evidence from a high-fat diet (HFD) xenograft model demonstrated that metformin significantly downregulated cyclin D1 and upregulated caspase 3 [80].

Several signaling pathways are affected by metformin exerting their influence on HCC proliferation and apoptosis. The Sonic hedgehog (Shh) signaling pathway plays a critical role in embryonic development, but aberrant activation of this pathway has been linked to the development of several types of cancer, including HCC [81]. Hu et al., using recombinant human Shh in HepG2 cells, evaluated the effects of metformin on proliferation in vitro. They documented that metformin inhibited Shh-induced proliferation. In addition, the expression of the mRNA and protein components of the Shh pathway, such as *Shh*, *Ptch*, *Smo* and *Gli-1*, decreased with metformin. They demonstrated that metformin exerts its inhibitory effects on the Shh pathway via AMPK, as the silencing of *AMPK* in the presence of metformin abrogated its inhibitory effects [82]. Zhao et al. investigated the effects of metformin in suppressing IL-22-induced HCC, which is known to promote HCC by activating the STAT3 signaling pathway. They showed that metformin inhibited the proliferation of SMMC-7721 and 97H cells induced by IL-22, and this was associated with the upregulation of the Hippo signaling pathway. Metformin upregulated the expression of several key components of the Hippo signaling pathway, including MST1, MST2, LATS1 and YAP, and downregulated the expression of the oncogenic protein c-Myc [83]. Conclusively, they suggested that the Hippo pathway might mediate the antitumor effects of metformin in HCC. In the same way, Vacante et al. suggested that metformin mediates its antitumor effects through multiple mechanisms. They documented that metformin upregulated KLF6 expression, which, in turn, led to the upregulation of p21, a cyclin-dependent kinase inhibitor that induces cell cycle arrest and inhibits cell proliferation. By promoting KLF6/p21-mediated cell cycle arrest, metformin could inhibit the growth and proliferation of HCC cells. Another mechanism involves the downregulation of the insulin-like growth factor (IGF) axis, which is known to promote cell proliferation and tumor growth [84]. Specifically, metformin downregulated the expression of *IGF-IR*, *IGF-II* and *IGF-IIR* [84]. Another pathogenetic mechanism is the generation of tumor-promoting isoforms. Zhuo et al. demonstrated that LGR4 expression was higher in the Huh-7 and HepG3B cell lines and in tumor tissues. The overexpression of LGR4 resulted in an increase in cell proliferation that was reduced by metformin [85]. Additionally, they documented that metformin induced switching properties in LGR4 isoforms that, collectively, support the hypothesis that metformin regulates HCC development by altering the alternative splicing of LGR4 [85]. As mentioned above, the fatty acid synthesis pathway is subjected to modifications upon metformin treatment [78]. Wang et al. proposed that the absence of fatty acid transport protein-5 (FATP5) led to a reprogramming of cellular energy metabolism, causing the liver cells to switch to a more glycolytic form of energy production. In detail, the knockdown of *FATP5* increased the cellular glycolytic flux and ATP production, thereby inhibiting AMPK and activating its downstream signaling mTOR to support HCC proliferation [86]. Finally, Cheng et al. provided evidence for the involvement of *SLC25A47*, a hepatic mitochondrial NAD+ transporter, in the regulation of the NAD+ levels, AMPKα activity and lipid metabolism in HCC cells. Specifically, knocking down *SLC25A47* reduced the levels of NAD+ in the mitochondria and decreased the activity of AMPKα, resulting in an accumulation of fatty acids and triglycerides in liver cells. They concluded that a *SLC25A47* knockdown reduced tumor growth in vivo in a diethyl nitrosamine (DEN)-induced mouse model. According to Cheng et al., *SLC25A47* is a therapeutic target of metformin and could be utilized as a treatment option in HCC [87].

It is worth mentioning that mounting evidence suggests that metformin could synergize with other molecules or modalities to enhance their cytotoxic capacity. Several studies have shown that combining metformin with other drugs can increase its antitumor effects in HCC, such as sodium glucose co-transporter-2 (SGLT-2) inhibitor empagliflozin [88], aloin [89], antifolates [90], dichloroacetate (DCA) [91], celastrol [92] and systemic hypoxia [93,94]. Zhang et al. developed a new pH-responding magnetic nanocomposite based on reduced graphene oxide that combines cisplatin and metformin to kill HepG2 and Caco-2 cells. The nanocomposite was created by linking polyhydroxyethyl methacrylic (PHEA) to reduced graphene oxide using a grafting-from approach via ATRP polymerization (Fe3O4@rGO-G-PSEA). They documented that the nanoparticles loaded with metformin/cisplatin exhibited the lowest concentration rate of HepG2 and Caco-2 cells compared to the drug-loaded single nanocomposite groups and free drugs by promoting an apoptotic response. Moreover, Fe3O4@rGO-G-PSEA demonstrated robust effectiveness against tumors in vivo while causing minimal harm to healthy tissues [95].

### 4.2. Metformin in Potentiation of Sorafenib

Sorafenib has been approved for the treatment of advanced HCC, and it has been the mainstay of medical treatment during the last decade [96]. Several clinical trials have demonstrated the efficacy of sorafenib in the treatment of advanced HCC. In particular, the Sorafenib HCC Assessment Randomized Protocol (SHARP) trial showed that sorafenib improved the overall survival in patients with advanced HCC compared to a placebo [97], as also validated by the Asia-Pacific trial. The latter demonstrated that sorafenib significantly improved the overall survival, time to symptomatic progression and time to progression in patients with advanced HCC in the Asia-Pacific region [98]. Evidence suggests that metformin could enhance sorafenib cytotoxicity.

Harati et al. used the HepG2 and Huh7 cell lines to investigate the effects of metformin, sorafenib and the combination of both drugs on cell proliferation and apoptosis. They showed that the combination of metformin and sorafenib had a synergistic effect on cell proliferation inhibition and apoptosis in HepG2 cells. However, in Huh7 cells, the combination of the two drugs did not show a significant synergistic effect on cell proliferation, inhibition or apoptosis [99]. Various mechanisms have been proposed for the above effects. Huang et al. demonstrated that the combination of phenformin and sorafenib significantly inhibited liver cancer cell proliferation and induced apoptosis compared to either drug alone. The combination treatment also led to the downregulation of the CRAF/ERK and PI3K/AKT/mTOR pathways [100]. Siddharth et al. further demonstrated that metformin and sorafenib led to a significant inhibition of HCC cell growth and induced apoptosis compared to either drug alone, suppressing the MAPK/ERK/Stat3 signaling pathway [101]. Another implicated pathway is the mTOR signaling pathway. Ling et al. evaluated the status of p-mTOR, p-AKT in the xenograft tumors following treatments with metformin, sorafenib and their combination. The results were consistent with the in vitro findings, with sorafenib significantly reversing the activation status of mTORC2 and the combined use of metformin and sorafenib synergistically suppressing mTORC1 activity [102]. Tang et al. evaluated the effects of sorafenib plus metformin on ferroptosis. They showed that the combination of metformin and sorafenib induced ferroptosis in HepG2 and HUH-7 cells, which was accompanied by increased lipid peroxidation and decreased cell viability compared to either drug alone. The combination treatment also led to suppression of the p62/Keap1/Nrf2 pathway, which is known to regulate cellular responses to oxidative stress [103]. Ren et al. proposed another mechanism of metformin and sorafenib synergy. They documented that metformin can inhibit the CXCR3 signaling pathway, which has been linked to sorafenib resistance in HCC cells, promote sorafenib sensitivity in vitro modulating the AMPK signaling pathway and induce metabolic alterations in HCC cells, such as decreasing the glucose uptake and lactate production, which may contribute to its antitumor effects and ability to sensitize HCC cells to sorafenib [104]. However, further studies are needed to validate these findings and evaluate the clinical relevance of these mechanisms in HCC patients.

### 4.3. The Effects of Metformin on HCC Autophagy

Autophagy is a cellular process by which cells degrade and recycle their own components, including proteins, organelles and other cellular debris. This process is essential for maintaining cellular homeostasis and responding to cellular stress [105]. Autophagy can have a tumor-suppressive effect by promoting the degradation of damaged or dysfunctional organelles and proteins that could otherwise promote HCC development. On the other hand, autophagy can also have a tumor-promoting effect by providing nutrients and energy to cancer cells and allowing them to survive in nutrient-poor conditions. This may contribute to the development of drug resistance and tumor recurrence in HCC [105]. Evidence suggests that metformin exerts a multitude of effects in HCC progression regulating autophagy.

Song et al. utilized data from The Cancer Genome Atlas (TCGA) and Gene Expression Omnibus (GEO) databases to identify the signatures of autophagy-related gene pairs (ARGP) that were significantly associated with the survival of HCC patients. They demonstrated that six ARGP signatures consisting of *BAK1/PELP1*, *BIRC5/CDKN2A*, *BIRC5/RGS19*, *CAPN2/ULK3*, *DIRAS3/TMEM74* and *PRKCD/RB1CC1* could categorize HCC patients into two distinct subgroups, demonstrating a significant difference in the overall survival (OS) [106]. As regards the implicated molecular mechanisms, Lai et al. showed that metformin treatment alone had minimal effects on the viability of sorafenib-resistant HCC cells, but it significantly enhanced the antitumor effects of sorafenib treatment. They documented that metformin induced autophagy in sorafenib-resistant HCC cells, activating the AMPK signaling pathway [107]. Gao et al. investigated further the involved molecular mechanisms. They showed that metformin induced autophagy in the MHCC97H and HepG2 cell lines, as evidenced by an increase in the number of autophagic vacuoles and the conversion of LC3-I to LC3-II. They concluded that metformin-induced autophagy was dependent on the activation of the AMPK signaling pathway and the inhibition of mTOR signaling [108]. However, further studies are needed to validate these findings in vivo and to assess the potential toxicity and long-term effects of metformin-induced autophagy in HCC.

### 4.4. The Effects of Metformin in HCC Metastasis

The molecular mechanisms of metastasis in HCC are complex and involve multiple processes such as epithelial–mesenchymal transition (EMT), by which epithelial cells lose their polarity and acquire a mesenchymal phenotype, allowing them to migrate and invade the surrounding tissues, angiogenesis [109], cell adhesion, proteases and immune evasion [110]. A recent study provided evidence that a five-gene signature related to EMT (*P3H1*, *SPP1*, *MMP1*, *LGALS1* and *ITGB5)* could be a useful prognostic tool for HCC patients. The authors developed a risk score formula based on the expression levels of these five genes, which was used to classify patients into low-risk and high-risk groups. They found that patients in the high-risk group had significantly worse overall survival and disease-free survival rates compared to those in the low-risk group [111]. Analogously, Zhu et al. demonstrated that a set of five EMT-related genes (*COL11A1*, *FN1*, *LAMC2*, *POSTN* and *THBS2*) were consistently upregulated in HCC samples and were associated with a poor patient prognosis. They found that the signature was an independent predictor of patient survival and could predict survival better than the clinical factors alone [112]. Mounting evidence links metformin with the regulation of multiple of these processes, which might have beneficial outcomes for HCC patients [80].

Regarding the effects of metformin on EMT and angiogenesis, tons of evidence have started to emerge. As mentioned above, Wang et al. reported that FATP5 deficiency led to a metabolic shift in HCC cells characterized by an increased glucose uptake, lactate production and ATP generation through glycolysis, as well as decreased mitochondrial respiration and fatty acid oxidation. The low expression of FATP5 in HCC tissues was associated with aggressive and invasive clinical and pathological characteristics. They reported that FATP5 inhibited EMT and suppressed the migration and invasion of SNU449 and MHCC97H cells. Conversely, when *FATP5* was silenced, it promoted a cellular glycolytic flux and ATP production, thereby suppressing the activation of AMPK and activating downstream signaling of mTOR, which supported HCC progression and metastasis. The metformin-induced activation of AMPK resulted in the reversal of EMT and reduced the metastatic ability of FATP5-depleted HCC cells [86]. Several cellular adhesion molecules and proteases regulate angiogenesis and the EMT process. Abdelhamid et al. reported that the diethylnitrosamine-induced HCC animal model group had an increased NF-κB level, which led to changes in EMT and angiogenesis. This increase in angiogenesis, metastasis and EMT was indicated by elevated levels of MMP-2/TIMP-1 and VEGF. However, treatments with metformin, empagliflozin or their combination reduced the VEGF and MMP-2/TIMP-1 ratio, which was shown to decrease angiogenesis and metastasis [88]. Finally, the increased matrix stiffness attenuates the inhibitory effect of metformin on HCC invasion and metastasis. Gao et al. demonstrated that HCC cells grown on a higher-stiffness substrate showed obvious resistance to the modulatory effects of metformin on migration, invasion and metastasis compared to the controls on lower-stiffness substrates. High stiffness induced an integrinβ1-mediated activation of the miR-17-5p/PTEN/PI3K/Akt signaling pathway, resulting in the upregulation of MMP2 and MMP9 expression. For the same-stiffness substrate, metformin upregulated PTEN expression and suppressed the activation of the PI3K/Akt/MMP pathway but had no effect on integrin β1 expression [113].

### 4.5. Metformin and Epigenetic Regulation of HCC

Epigenetic modifications are alterations to DNA and chromatin structures that affect gene expression without changing the underlying DNA sequence. Epigenetic alterations that play a role in HCC are DNA methylation, where a methyl group is added to a cytosine base in DNA, histone modifications that alter the chromatin structure and affect gene expression, noncoding RNA and chromatin remodeling [114]. Recent evidence has provided important insights into the mechanisms by which metformin alters DNA methylation [115] and regulates the expression of noncoding RNAs [116], with implications for the use of metformin in cancer therapy.

Zhong et al. explored the mechanism by which metformin alters DNA methylation via the H19/S-adenosylhomocysteine hydrolase (SAHH) axis [115]. They demonstrated that metformin led to global changes in DNA methylation in cancer cells, as evidenced by reduced representation bisulfite sequencing (RRBS). This was mediated by activating AMPK, which, in turn, induced the upregulation of microRNA let-7. This microRNA degraded H19 long noncoding RNA, which usually binds and inhibits SAHH. By knocking down H19, SAHH is activated, and DNA methyltransferase 3B(DNMT3B) can then methylate a subset of genes. These findings demonstrate a previously unknown mechanism of action for metformin, which has implications for the molecular mechanisms of epigenetic dysregulation in cancer [115]. Peng et al. presented a study that aimed to investigate the role of plasma exosomal miR-122 in the regulation of the efficacy of metformin. MiR-122 is predominantly expressed in the liver and plays a crucial role in the regulation of hepatic lipid metabolism, inhibiting the phosphorylation of AMPK. They found that the levels of exosomal miR-122 were significantly lower in patients with T2D and HCC compared to healthy controls. Additionally, the levels of exosomal miR-122 were positively correlated with the efficacy of metformin in T2D patients. They documented that the expression of miR-122 enhanced the sensitivity of HepG2 cells to metformin, whereas decreased miR-122 expression resulted in hepatocyte insensitivity to metformin. As a result, significantly elevated levels of miR-122 in plasma exosomes of hepatocellular carcinoma patients could improve their sensitivity to metformin. These findings revealed a crucial regulatory role of plasma exosomal miR-122 in the molecular mechanism of metformin. By regulating key molecules of related signaling pathways, miR-122 may produce similar therapeutic effects on glycemic control and tumor suppression as metformin. This offers new possibilities for developing therapeutic strategies for hepatocellular carcinoma based on miR-122 and metformin [116].

Overall, while the preclinical evidence is promising, more research is needed to confirm the potential of metformin in targeting these cancer hallmarks and to determine its clinical effectiveness in the prevention and treatment of HCC and other types of cancer.

All the above effects are summarized in Figure 2.

## 5. Metformin as an Immunotherapy Enhancer

While immunotherapy has become the mainstay of treatment in HCC, there is currently no established biomarker that can predict patient responses or resistance to immune checkpoint inhibitors (ICIs) [7,117]. Inflammation is a key factor in the pathogenesis and progression of HCC, and an increased neutrophil to lymphocyte ratio (NLR ≥ 5) and absolute neutrophil count have been linked to advanced disease, poor prognosis and poor response to various HCC treatments, including ICIs [118]. Recent studies have suggested that NASH-related HCC may be less responsive to immunotherapy due to the presence of CD8+ T cells, particularly the CXCR6+ subset induced by hepatic steatosis, which can cause hepatocyte injury and inflammation through the secretion of proinflammatory cytokines and tumor necrosis factor (TNF) [119]. Additionally, NASH diminishes the effectiveness of immunotherapeutic agents such as M30 and aOX40, reducing the infiltration of CD4+ T cells and effector memory cells into the HCC tumor microenvironment (TME) [120]. In parallel, the contribution of innate immunity inflammatory cells such as TAMs [121] and TANs [122] and of signaling systems such as the NOD-like receptor family, pyrin domain-containing protein 3 (NLRP3) inflammasome [123] has been increasingly recognized for their contributions in the progression of HCC. Recent evidence has implicated metformin in the regulation of HCC immune TME [124]. In addition to the induction of apoptosis, which was analyzed above, metformin attenuates HCC cell proliferation, activating proptosis. This effect is partially dependent on FOXO3, which can activate the transcription of NLRP3 [125]. Additionally, AMPK upregulation or metformin use downregulates NF-kB signaling, activating IkBa [75]. Summarizing the evidence in the literature, metformin can promote HCC immunotherapy by enhancing the anti-PD-1 efficacy, potentiating the reshape of the immune TME and downregulating the pro-metastatic capacity of neutrophil extracellular traps (NETs) which, will be described below in depth.

### 5.1. Enhancing the Efficacy of Anti-PD-1

In a previous study by Sia et al. in 2017, an immune classification of HCC was documented. Twenty-five percent of HCCs expressing PD-1, PD-L1 and molecules related to cytolytic activity were identified as the “immune class”. The “immune class” was further subcategorized into an “immune active” subclass with favorable prognosis that expressed molecules of the adaptive T-cell response and an “immune exhausted” subclass characterized by the elevated expression of several genes regulated by transforming growth factor beta 1 (TGF-b1) that mediate immunosuppression [126]. Montironi et al. took it a step further, recognizing an “immune-like subclass” that was analogous with the “immune class” but exhibited significant activation of the Wnt/β-catenin pathway, which was attributed to *CTNNB1* mutations. This subtype was not captured by the previous immune signature. The inflamed class proposed by Montironi et al., which constitutes approximately 35% of HCC cases, displays the activation of interferon and PD-1 signaling and upregulation of the genes associated with lymphocyte chemotaxis, including *CXCL9* and *CXCL10* [127]. Wabitsch et al. took it a step further, demonstrating that, besides the infiltration or the exhaustion state, tumor immune cells in HCC TME exhibit motility disorders. In order to study NASH-driven HCC and its resistance to anti-PD-1 treatment [119], they used a collection of mouse models that were fed various diets and transplanted with liver tumor or liver metastatic cells. Mice with NASH-HCC did not respond well to anti-PD-1 treatment, while mice with HCC but without NASH showed significant tumor reduction. To understand the reasons behind this difference in response, Wabitsch et al. analyzed CD8+ T lymphocytes in the livers of mice with NASH. They found that mice with NASH had a high infiltration of effector CD8+ T cells in their livers. However, when CD8+ T-cell infiltration was specifically analyzed in the tumor, the levels observed in mice with or without NASH were comparable [128]. The tumor microenvironmental dynamics in vivo were subsequently explored by the authors by performing intravital imaging. They found that intra-tumoral CD8+ T cells in mice with NASH had a lower speed and shorter displacement length compared to those in mice without NASH. The researchers also discovered that CD8+ T cells from NASH-HCC livers had alterations in multiple metabolic pathways, including glycolysis, fatty acid oxidation and mitochondrial respiration, which were associated with mitochondrial depolarization and a loss of mitochondrial mass. To further elucidate the impaired motility of CD8+ T cells, they performed a transcriptomic analysis focused on metabolic genes. They found that metformin reprogramed CD8+ T cells and restored their motility and sensitivity to immunotherapies. Finally, they demonstrated the translational potential of metformin, as it synergized with anti-PD-1 monotherapy or with the combination immunotherapy of anti-VEGFR2 and anti-PD-L1 [124,128]. For additional study into the T-cell motility dynamics, an extensive review has been conducted elsewhere [129].

The process of folate metabolism, which encompasses a variety of biochemical reactions collectively known as one-carbon (1C) metabolism, plays a fundamental role in activating and transferring 1C molecules for use in biosynthesis [130]. This universal metabolic process produces essential compounds such as purine, thymidine, methionine and nicotinamide adenine dinucleotide phosphate (NADPH), which are critical for DNA synthesis, epigenetics and maintaining the mitochondrial redox homeostasis [130]. Recent research on metabolism has shown that there is competition for glucose and methionine between tumors and T cells, which can impact the activity of TILs within the tumor. Peng et al. demonstrated that phosphoserine phosphatase (PSPH), an upstream component of the 1C metabolic pathway, may have a significant impact on the immune formation of HCC TME [131]. They demonstrated a new function of PSPH in 1C metabolism that aids in the progression of HCC by creating a TME with high levels of myeloid cells and low levels of effector T cells. They found that PSPH has varying effects on the expression of CCL2 and CXCL10 in cancer cells. The SAM pathway, which is downstream of the 1C pathway, was responsible for the downregulation of C–X–C motif chemokine ligand 10 (CXCL10) through H3K27me3-mediated gene expression regulation, while GSH mediated the upregulation of C–C motif chemokine ligand 2 (CCL2), neutralizing the ROS and thereby activating the STAT3 signaling pathway. Importantly, when PSPH was downregulated, the effectiveness of anti-PD-1 therapy was significantly improved in mice, indicating that PSPH is not only a biomarker but also a promising drug target for future combination therapies [131]. It is important to note that there is currently no commercially available inhibitor that selectively targets PSPH. Thus, they found that metformin could imitate the effects of *shPSPH*, reducing PSPH expression in HCC cells. This reduction led to a decrease in the CCL2 levels and macrophage infiltration while increasing the CXCL10 levels and the CD8+ T-cell infiltration in tumor tissues. As a result, metformin inhibited tumor growth in mice in vivo. Notably, a combination of metformin and anti-PD-1 antibodies showed superior antitumor effects compared to either treatment alone. This finding highlights another similarity between the effects of metformin and *shPSPH*. It is essential to recognize that the study provided evidence for the potential repurposed use of metformin in treating HCC and offered additional evidence for the role of PSPH in modulating the immune TME of HCC [131]. The above are illustrated in Figure 3.

### 5.2. Reversing of Effector T-Cell Exhaustion State

It is well known that heterozygous deletion of the *Ncoa5* gene, which encodes the nuclear receptor coactivator 5 protein, results spontaneously in HCC. The development of tumors is preceded by the increased expression of interleukin-6 (IL-6), early-onset glucose intolerance, progressive steatosis and liver dysplasia [132]. Chronic inflammation can lead to tumorigenesis through various mechanisms, including the creation of an immunosuppressive environment. *Ncoa5+/−* male mice with preneoplastic livers were found to have an immune environment that contained increased populations of activated and tissue-resident memory (TRM) CD8+ T lymphocytes and immunosuppressive cells such as myeloid-derived suppressor cells (MDSC) and M2 macrophages [133]. While activated and memory CD8+ T cells play a crucial role in cancer surveillance and have positive prognostic values, the continuous growth and recurrence of HCCs indicate the failure of the effective immune control of cancer progression. One of the reasons for this is the existence of various immunosuppressive mechanisms in the protumorigenic microenvironment, including CD8+ T-cell exhaustion and dysfunction due to chronic inflammation [134]. Metformin has been found to inhibit the immune exhaustion of tumor-infiltrating CD8+ lymphocytes in multiple types of established cancers in mice, suggesting that it may be useful in reversing the immunosuppressive and CD8+ T-cell-exhausted tumor microenvironment in HCC patients. Williams et al. suggested that metformin might be able to alleviate this situation, reducing the hepatic infiltration of CD8+ T cells, macrophages and the enrichment of exhausted CD8+ and Stat3 signaling gene signatures in *Ncoa5+/−* male mice. Thus, our data, combined with the mechanistic insights into metformin’s actions, could be valuable in supporting the potential benefit of metformin treatment in reversing the immunosuppressive and CD8+ T-cell-exhausted tumor microenvironment in HCC patients [133]. The above evidence needs further evaluation in clinical trials in order to evaluate their relevance in clinical practice.

### 5.3. Metformin Effects on HCC Prevention/Reduction of Metastatic Potential

Neutrophilic extracellular traps (NETs) are web-like structures composed of DNA and various proteins, including histones, neutrophil elastase, myeloperoxidase and cathepsin G, that are released by neutrophils in response to infections, inflammation and tissue damage [135]. Recent evidence links NET with HCC progression and metastasis [136]. Wang et al. demonstrated that, in NASH, although the total count of CD4+ T cells was lower, there was a specific increase in the Treg subpopulation. The inhibition of HCC initiation and progression in NASH was significantly reduced upon the depletion of Tregs. An increase in the hepatic Treg levels is positively associated with an elevated NET. RNA sequencing data revealed that NETs had an impact on the gene expression profiles of naive CD4+ T cells, since the most significantly altered genes were those involved in mitochondrial oxidative phosphorylation. The promotion of Treg differentiation can be facilitated by NETs through the facilitation of mitochondrial respiration. TLR4 is required for the metabolic reprogramming of naive CD4+ T cells by NETs. The activity of Tregs can be reduced by an in vivo blockade of NETs using *Pad4-/-* mice or DNase I treatment [137]. Jiang et al. documented that NETs play a pro-metastatic role in the environment of human HCC. They reported that the metabolic switch induced by tumors towards glycolysis and the pentose phosphate pathway in tumor-infiltrating neutrophils promoted the formation of NETs in a reactive oxygen species-dependent manner. Subsequently, NETs promoted the migration of cancer cells and downregulated tight junction molecules on adjacent endothelial cells, thus facilitating tumor extravasation and metastasis. The depletion of NETs inhibited tumor metastasis in mice in vivo, and the level of NET-releasing neutrophil infiltration was negatively associated with patient survival and positively correlated with the tumor metastasis potential of HCC patients [138].

Given the above, Yang et al. demonstrated that metformin could potentially target oxidized mitochondrial DNA (mtDNA) in NETs as an antimetastatic strategy [139]. Mitochondria generate reactive oxygen species, or mitoROS, which play several important roles in energy metabolism and maintaining the cellular balance. MtDNA is a circular double-stranded DNA molecule that is more susceptible to oxidative stress compared to nuclear DNA (nDNA). Additionally, mtDNA is highly proinflammatory and a more potent TLR ligand than nDNA [140]. The generation of mitoROS in neutrophils is critical for the formation of NETs. Yang et al. found high levels of mitoROS in neutrophils of HCC patients, leading to the formation of NETs that contained oxidized mtDNA in a mitoROS-dependent manner. The presence of NETs and oxidized mtDNA was clinically significant. When bound to NET proteins, oxidized mtDNA was more effective in inducing metastasis-promoting inflammatory mediators in HepG2 cells. Targeting oxidized mtDNA with metformin, the metastasis-promoting inflammatory state was reduced, thus weakening the metastatic capacity of HCC [139]. Overall, while the study provided interesting and potentially important findings, further studies are needed to confirm the role of NETs and oxidized mtDNA in HCC and to determine their potential as therapeutic targets in the management of metastatic disease.

## 6. Discussion

Despite the initial enthusiasm about the introduction of immunotherapy in clinical practice, ICIs are accompanied by a multitude of toxicities [141]. Rimassa et al., in a state-of-the-art review, thoroughly summarized the current evidence about immunotherapy in HCC [142]. Advanced HCC patients have been treated with single-agent ICIs, but the objective response rate was only 15–20% [143], and the OS was not significantly improved. Moreover, about 30% of HCC cases demonstrated an inherent resistance to ICIs [144]. Lacking biomarkers to predict which patients are most likely to benefit from immunotherapy, researchers have begun to explore combinations of ICIs with potential efficacy in broader patient populations. To this end, basket trials with HCC cohorts and early-phase studies have investigated combining ICIs with antiangiogenic agents or with two different ICIs. These initial studies showed promising results, which led to the conduct of phase 3 trials that tested the combination of anti-PD-1/PD-L1 with bevacizumab [145], tyrosine kinase inhibitors (TKIs) or anti-CTLA-4. The IMbrave150 trial demonstrated that the combination of atezolizumab-bevacizumab was the first regimen to show improved survival in a front-line setting since sorafenib’s approval [145]. Similarly, the superiority of durvalumab-tremelimumab (STRIDE regimen) over sorafenib was demonstrated by the HIMALAYA trial [146]. However, the results of combining ICIs and TKIs were inconsistent, with only one phase 3 trial showing an OS benefit [147]. Despite the rapidly evolving therapeutic landscape for advanced HCC patients, there are still areas that necessitate further investigation. These include identifying the best treatment choices and sequences, discovering biomarkers, exploring combinations with locoregional therapies and creating new immunotherapy agents [142].

The results of late-phase efficacy studies that evaluated metformin as a cancer therapeutic after repurposing it have not been satisfactory. However, there are still promising avenues for research in specific populations, including exploring the combination of metformin with immunotherapy and investigating its potential as a cancer-preventative agent [148]. The activation of AMPK in immune cells results in the phosphorylation of PD-L1, followed by its glycosylation and accumulation within the endoplasmic reticulum, ultimately leading to degradation [149]. Li et al. reported that AMPK activation induced by metformin may lead to the downregulation of CD39 and CD79 gene expression, which, in turn, can reduce the immunosuppression driven by MDSCs [150]. Additionally, preclinical investigations have shown that metformin can alter the polarization of macrophages from a M2- to a M1-like phenotype, which can inhibit tumor growth and angiogenesis. This effect may be driven by the activation of AMPK/NF-κB signaling [151].

Metformin has been studied extensively for its potential anticancer properties in HCC, and several pros and cons have been documented. With regards to the advantages, firstly, preclinical studies have shown that metformin has multilayer antitumor effects on HCC through various mechanisms, including the inhibition of cancer cell proliferation, induction of apoptosis and suppression of cancer cell invasion and migration. Secondly, observational studies on humans have suggested that metformin use may be associated with a reduced risk of HCC incidence and mortality. Thirdly, metformin has a good safety profile, with relatively few side effects, making it an attractive candidate for use in cancer treatment. On the other hand, firstly, there is a lack of randomized controlled trials (RCTs) investigating the efficacy of metformin on HCC. Secondly, most studies have been observational and may have included biases and confounding factors, and thirdly, the optimal dose and duration of metformin treatment for HCC are still unclear, while the doses that have been used in preclinical studies may not be achievable in humans due to safety concerns. Lastly, the heterogeneity of HCC, including the underlying etiologies and stages of the disease, may limit the generalizability of the findings from studies investigating metformin in this context.

## 7. Conclusions

Overall, while there is some promising evidence suggesting that metformin may have potential as a therapeutic option for HCC, further research in the form of well-designed RCTs is essential in order to determine its true efficacy, optimal dosing and duration and safety profile in this context. While mounting evidence suggests that metformin might enhance the activity of immune cells, more research is needed to shed light into the exact mechanisms by which metformin might enhance immunotherapy in HCC. In the era of personalized medicine and cutting-edge individualized nanotherapy [152], the concept of repurposing a drug from the World Health Organization’s list of essential medicines to enhance immunotherapy responses can be likened to discovering the “Achilles heel” of HCC. Repurposing drugs may provide a cost-effective and expedient alternative to developing new therapeutics [148]. This innovative approach has the potential to improve patient outcomes and contribute to the ongoing effort to enhance global health.

## Figures and Tables

**Figure 1 cancers-15-03161-f001:**
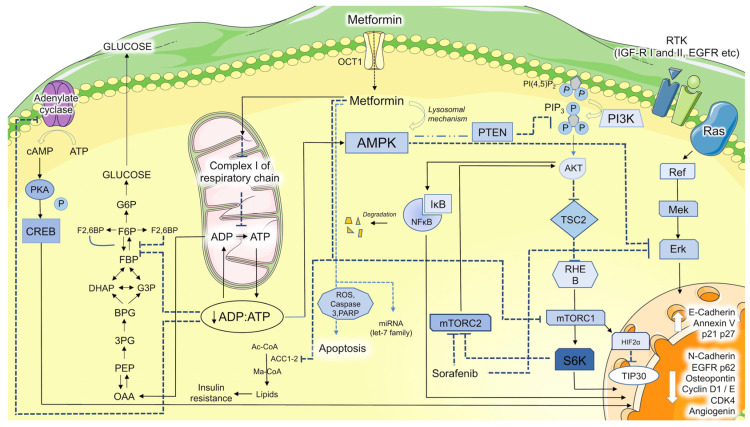
**Metformin’s mechanisms of action on the metabolism and intracellular signaling pathways.** *OCT1, organic cation transporter 1; RTK, receptor tyrosine kinases; PI3K, phosphatidylinositol 3-kinase; PI(4,5)P2, phosphatidylinositol-4,5-diphosphate; PIP3, phosphatidylinositol-3,4,5-trisphosphate; AMPK, AMP-activated protein kinase; cAMP, cyclic adenylate mono-phosphate; ADP, adenylate di-phosphate; ATP, adenylate tri- phosphate; PKA, protein kinase A; CREB, cAMP response element-binding protein; G6P, glucose-6- phosphate; F6P, fructose-6-phosphate; F2,6BP, fructose-2,6-biphosphate; FBP, fructose-1,6-biphosphate; DHAP, dihydroxyacetone phosphate; G3P, glyceraldehyde 3-phosphate; BPG, 1,3-bisphosphoglycerate; 3PG, 3- phosphoglycerate; PEP, phosphoenolpyruvate; OAA, oxaloacetate; Ac-CoA, acetyl-CoA; Ma-CoA, malonyl-CoA; ACC, Acetyl-CoA carboxylase; ROS, reactive species of oxygen; NF-κB, nuclear factor kappa-light-chain-enhancer of activated B cells; IκB, inhibitors of κB; mTORC, mammalian target of rapamycin complex; HIF2α, hypoxia induced factor 2α*.

**Figure 2 cancers-15-03161-f002:**
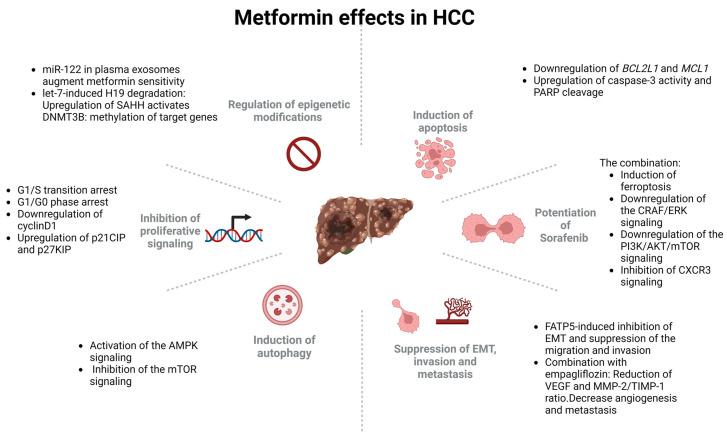
Summarizes the effects of metformin in HCC. Created with BioRender.com.

**Figure 3 cancers-15-03161-f003:**
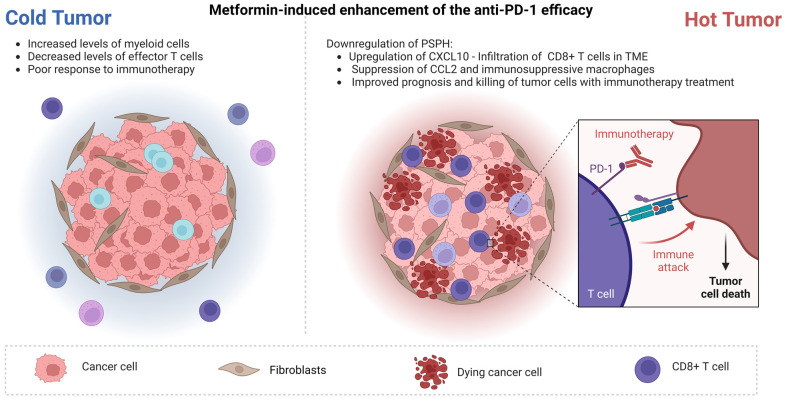
The effects of metformin in reshaping the HCC immune TME. Created with BioRender.com.

**Table 1 cancers-15-03161-t001:** Outlines the meta-analyses about the effects of metformin in HCC.

Author; Year	Methods	Outcomes	Ref
Zhou; 2020	Six retrospective cohort studies: four studies involving curative treatments for HCC and two studies involving noncurative treatments. The curative treatment studies: 618 patients on metformin and 532 patients on other antihyperglycemic agents, while the noncurative treatment studies: 92 patients on metformin and 57 patients on other antihyperglycemic agents.	Patients treated with metformin had significantly longer OS rates compared to those treated with other antihyperglycemic agents after curative therapies. The odds ratios (ORs) for one-year, three-year and five-year OS were 2.62 (95% confidence interval (CI): 1.76–3.90), 3.14 (95% CI: 2.33–4.24) and 3.31 (95% CI: 2.39–4.59). Similarly, metformin treatment was associated with improved recurrence-free survival (RFS) rates after curative therapies. The OR for one-year and three-year RFS were 2.52 (95% CI: 1.84–3.44) and 2.87 (95% CI: 2.15–3.84).	[20]
Li; 2022	Nine case-control studies (248,433 participants) and fifteen cohort studies (1,203,832 participants) to investigate the influence of metformin usage on the risk of HCC in DM patients. Nine studies (11,375 participants) examined the impact of metformin usage on the mortality of HCC in DM patients.	Metformin was associated with a reduced risk of HCC in DM patients OR/RR 0.59 (95% CI: 0.51–0.68) with a high degree of heterogeneity (I2 = 96.5%). Metformin usage was linked to a decreased risk of all-cause mortality in individuals with DM who had been diagnosed with HCC HR 0.74 (95% CI: 0.66–0.83) with moderate heterogeneity (I2 = 49.6%).	[59]
Memel; 2021	The analysis encompassed a total of 2,389,019 participants (20,479 incident HCC cases). They assessed the combined RRs and corresponding 95% CIs to assess the association between aspirin usage and the risk of incident HCC.	The beneficial effect of aspirin was notably stronger in studies that accounted for the concurrent use of statins and/or metformin (RR = 0.45, 95% CI: 0.28–0.64) compared to studies that did not consider these factors.	[57]
Yuan; 2022	The effectiveness of metformin in improving the OS and RFS among HCC patients diagnosed with T2DM following curative treatment: six studies—5936 patients.	The utilization of metformin was associated with significant improvements in the 3-year (OR = 1.50, 95% CI: 1.22–1.83) and 5-year (OR = 1.88, 95% CI: 1.47–2.41) OS rates. Metformin usage was linked to reduced rates of recurrence at 1-year (OR = 1.31, 95% CI: 1.08–1.59,), 3-year (OR = 1.88, 95% CI: 1.48–2.37) and 5-year (OR = 1.83, 95% CI: 1.40–2.40) intervals.	[56]
Campbell; 2021	The aim was to evaluate the risk factors contributing to the progression of HCC in individuals with chronic hepatitis B virus (HBV) infection: 68 studies, 25,447 cases of HCC and 576,792 patients with chronic HBV.	In studies that accounted for the use of metformin, the relationship between DM and an increased risk of HCC became less pronounced. In the analysis that was limited to adjusted studies for metformin, DM participants had a 16% higher risk of HCC compared to non-DM individuals (HR 1.16, 95% CI: 1.04–1.29).	[58]
Zeng; 2022	The association between metformin and the risk of HCC: 3 studies, 125,458 patients.	The utilization of metformin did not demonstrate an association with a reduced overall risk of HCC (HR: 0.57, 95% CI: 0.31–1.06).	[55]

**Table 2 cancers-15-03161-t002:** Latest clinical studies about the effects of metformin in patients with DM, NAFLD or viral hepatitis.

Author; Year	Country	Type of Study	Groups/N of Patients	Outcome	Ref
Antwi; 2019	USA	Retrospective study	Nonusers (N = 1193) Statins Only (N = 582) Metformin Only (N = 295) Both (N = 429)	A 28% lower risk of death after HCC diagnosis (HR, 0.72; 95% CI: 0.58–0.91) associated with an average daily metformin dose of ≤ 1500 mg before diagnosis compared to nonusers.	[63]
HsiehI; 2020	Taiwan	Retrospective study	Control (N = 353), DM (N = 91), HTN (N = 184) and DM + HTN (N = 105)/metformin (N = 63), OHA (n = 104) and RI/NPH (n = 29)	Control group vs. metformin group (7.70 vs. 12.60 months, *p* = 0.011) control group vs. non-metformin oral hypoglycemic agents group (7.70 vs. 10.80 months, *p* = 0.016) control group vs. insulin glargine/NPH group (7.70 vs. 15.20 months, *p* = 0.026).	[64]
Azit; 2022	Malaysia	Retrospective study	Metformin (Ν = 130), insulin (Ν = 71), sulphonylureas (Ν = 90)	Patients on metformin had a 1.44-times greater risk of mortality (AHR = 1.44, 95% CI: 1.03–2.00).	[65]
Hydes; 2022	UK	Retrospective study	Diet (Ν = 91), metformin (Ν = 171) thiazolidinedione (Ν= 1), DPP4 inhibitor (Ν= 7), sulfonylureas (Ν = 117), insulin (N = 126)	In diabetic patients with HCC, metformin was linked to a better OS, as evidenced by a mean survival of 31 months vs. 24 months (*p* = 0.016) and a hazard ratio (HR) for death of 0.75 (*p* = 0.032).	[66]

OHA = oral hypoglycemic agents, RI = regular insulin and NPH = neutral protamine Hagedorn.

**Table 3 cancers-15-03161-t003:** Recent evidence about the effects of metformin in conjunction with other primary therapeutic interventions.

Author; Year	Country	Type of Study	Groups	N of Patients	Overall Survival (%)			Ref
					1-Year	3-Year	5-Year	
**Curative intent surgery**								
Cao; 2022	China	RSC	Antidiabetic treatment (metformin or insulin or both) vs. no treatment	292 vs. 106	82.7 vs. 80.1	65.6 vs. 58.7	46.4 vs. 29.2	[67]
Luo; 2019	China	RSC	Metformin vs. other	63 vs. 113	94 vs. 77	76 vs. 49	53 vs. 29	[68]
Kang; 2018	South Korea	RSC	Metformin vs. other	45 vs. 225	NA	97.8 vs. 89.5	83.2 vs. 67.8	[69]
Chan; 2017	Taiwan	ND	Metformin vs. other	1632 vs. 2978	Among DM, those on metformin had significantly improved RFS and OS.			[70]
					A reduction in the risk of HCC recurrence after liver resection was positively correlated with the amount and duration of metformin use.			
**RFA/SBRT/** **HFRT/** **Radioembolization/** **TACE**								
Chen; 2022	China	RSC	Metformin vs. other	39 matched	43 vs. 35 months			[71]
Jung; 2022	South Korea	RSC	Metformin vs. other	47 vs. 47	No difference.			[72]
Elsayed; 2021	USA	RSC	Metformin vs. other	19 vs. 93	NA	38.2 months vs. 40.3 months	NA	[73]
Chen; 2011	Taiwan	RSC	Metformin vs. other	21 vs. 32	95 vs. 74.5	69.2 vs. 44.8	60.5 vs. 26.2	[74]

## Data Availability

Not applicable.

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
