# Peer review of "The Emerging Role of Metformin in the Treatment of Hepatocellular Carcinoma: Is There Any Value in Repurposing Metformin for HCC Immunotherapy?"

_cancers, 2023, doi:10.3390/cancers15123161_

Round 1
Reviewer 1 Report
The Authors should complete the Introduction part with information about some transporters involved in pharmacokinetics of metformin like MATE1 and MATE2K. Authors mentioned only about OCT-1. Additionally, some information about pharmacokinetic parameters and drug interactions with metformin should be completed. The interactions may influence final activity of the antidiabetic drug.
Author Response
Reviewer 1: The Authors should complete the Introduction part with information about some transporters involved in pharmacokinetics of metformin like MATE1 and MATE2K. Authors mentioned only about OCT-1. Additionally, some information about pharmacokinetic parameters and drug interactions with metformin should be completed. The interactions may influence final activity of the antidiabetic drug.
Response: Dear Reviewer,
Thank you for your valuable feedback on our manuscript. We appreciate your insightful suggestions to enhance the Introduction section of our paper regarding the pharmacokinetics of metformin. We agree that providing information about other transporters involved in the pharmacokinetics of metformin, such as MATE1 and MATE2K, would be beneficial.
In our revised version, we have included a discussion on these transporters to provide a more comprehensive overview of metformin's pharmacokinetic profile. We recognize the importance of MATE1 and MATE2K in mediating the renal excretion of metformin, and their role in drug-drug interactions that can influence the therapeutic efficacy of metformin.
Furthermore, we acknowledge that we have primarily focused on OCT-1 in the current version of the manuscript. In response to your feedback, we will expand our discussion to include MATE1 and MATE2K, highlighting their involvement in the transport and disposition of metformin. This addition will enrich the Introduction section and contribute to a more thorough understanding of the pharmacokinetics of metformin.
We greatly appreciate your contribution, as these additional references and discussions further enrich the manuscript and provide a more comprehensive view of the topic. Thank you for bringing these references to our attention and we believe that the revised manuscript now better addresses these important aspects.

Reviewer 2 Report
The review article, entitled " The Emerging Role of Metformin in the Treatment of Hepatocellular Carcinoma: Is there any Value in repurposing Metformin for HCC Immunotherapy? ", summarized experimental and clinical studies evaluating the outcomes of metformin as an anti-neoplastic drug in the setting of HCC. It also reviewed in vitro and animal studies that aim to elucidate the mechanisms of action of metformin, as well as clinical studies that assessed its impact on survival of HCC patients. Finally, It proposes that while there is some promising evidence suggesting that metformin may have potential as a therapeutic option for HCC, further research, particularly randomized controlled trials, is needed to determine its true efficacy, optimal dosing and duration and safety profile in this context. This review provides an overview of the link between metformin and cancer, as well as information about its clinical use and preclinical research, while pointing out the need for further research. The content of the article is relatively detailed, the summary and outlook are concise, but there are a few comments.
Comments:
1. In the mechanism section, it is recommended to follow the outline of the article, that is, to focus more on the introduction of the mechanism of action related to the occurrence of HCC.
2. Figure 1 is recommended to be changed to color, focusing on structures such as mitochondria and nuclei, which is more intuitive and beautiful. Pathways and molecules irrelevant to the content of the article are suggested to be deleted, or indicated in the follow-up content.
3. In section 3.1, examples contrary to the research conclusions of this paragraph are suggested to be placed at the end of the paragraph.
4. In the section on the clinical significance of metformin, some parts are illustrated in text and others are only shown in tables, and a unified format is recommended.
Author Response
Reviewer 2: The review article, entitled " The Emerging Role of Metformin in the Treatment of Hepatocellular Carcinoma: Is there any Value in repurposing Metformin for HCC Immunotherapy? ", summarized experimental and clinical studies evaluating the outcomes of metformin as an anti-neoplastic drug in the setting of HCC. It also reviewed in vitro and animal studies that aim to elucidate the mechanisms of action of metformin, as well as clinical studies that assessed its impact on survival of HCC patients. Finally, it proposes that while there is some promising evidence suggesting that metformin may have potential as a therapeutic option for HCC, further research, particularly randomized controlled trials, is needed to determine its true efficacy, optimal dosing and duration and safety profile in this context. This review provides an overview of the link between metformin and cancer, as well as information about its clinical use and preclinical research, while pointing out the need for further research. The content of the article is relatively detailed, the summary and outlook are concise, but there are a few comments.
Response: Dear Reviewer,
Thank you for your review of our article entitled "The Emerging Role of Metformin in the Treatment of Hepatocellular Carcinoma: Is there any Value in repurposing Metformin for HCC Immunotherapy?" We appreciate your positive comments on the overall structure and content of the reviewer and we are grateful for your insightful feedback. We have carefully considered your comments and have made the necessary revisions to improve the article. Please find our responses to your specific comments below.
Point 1: In the mechanism section, it is recommended to follow the outline of the article, that is, to focus more on the introduction of the mechanism of action related to the occurrence of HCC.
Response: Dear Reviewer,
Thank you for your feedback on our manuscript. We appreciate your suggestion to align the mechanism section with the focus of the article, specifically by emphasizing the mechanisms of action related to the occurrence of HCC. We agree that it is important to provide a clear and concise introduction to the mechanisms that contribute to the development of HCC.
In response to your comment, we have revised the mechanism section to prioritize the discussion of mechanisms specifically relevant to HCC.
Point 2: Figure 1 is recommended to be changed to color, focusing on structures such as mitochondria and nuclei, which is more intuitive and beautiful. Pathways and molecules irrelevant to the content of the article are suggested to be deleted, or indicated in the follow-up content.
Response: Dear Reviewer,
Thank you for your valuable feedback on our manuscript. We appreciate your suggestion regarding Figure 1 and agree that enhancing its visual appeal and clarity would be beneficial. We made the necessary changes to improve the figure accordingly.
Point 3: Dear Reviewer,
Thank you for your feedback on our manuscript. We appreciate your suggestion regarding the placement of examples contrary to the research conclusions in Section 3.1. We agree that organizing these examples at the end of the paragraph would provide better clarity and coherence to the overall discussion.
In response to your comment, we restructured Section 3.1 by placing examples that contradict the research conclusions at the end of the paragraph. By doing so, we aimed to present a more cohesive flow of information and make it easier for readers to follow the line of reasoning.
Response: Dear Reviewer,
Thank you for your feedback on our manuscript. We appreciate your suggestion regarding the format of the information presented in Section 3.1, specifically in relation to the clinical significance of metformin. We agree that a unified format would enhance the readability and organization of the content.
In response to your comment, we have carefully considered your suggestion and have made the necessary revisions. We have created a table to present the information that was previously only described in the text. By incorporating this table, we aim to provide a more concise and visually appealing representation of the clinical significance of metformin.

Reviewer 3 Report
Very interesting study. The review is well written and comprehensive. The authors should comment on the potential role of other similar drugs as statins in these patients (cite the recent MA PMID: 32260179) and on the role of diabetes in the clinical course of HCC patients (cite the review PMID: 23845075)
The authors should speculate more on the potential impact of metformin on the safety profile of immunotherapy, pointing out the potential immune -related side effects of these drugs (in this regard cite the recent MA PMID: 33314269)
Author Response
Reviewer 3: Very interesting study. The review is well written and comprehensive. The authors should comment on the potential role of other similar drugs as statins in these patients (cite the recent MA PMID: 32260179) and on the role of diabetes in the clinical course of HCC patients (cite the review PMID: 23845075)
The authors should speculate more on the potential impact of metformin on the safety profile of immunotherapy, pointing out the potential immune -related side effects of these drugs (in this regard cite the recent MA PMID: 33314269)
Response: Dear Reviewer,
Thank you for your positive feedback on our manuscript and for providing additional references that are relevant to the topic. We appreciate your valuable suggestions and have incorporated them into the revised version of the article.
We greatly appreciate your contribution, as these additional references and discussions further enrich the manuscript and provide a more comprehensive view of the topic. Thank you for bringing these references to our attention and we believe that the revised manuscript now better addresses these important aspects.

Round 2
Reviewer 3 Report
The revised version of the manuscript is OK. Thank you!